# Dissolution of spiral wave's core using cardiac optogenetics

**Sayedeh Hussaini**[1,2,3]*, **Sarah L. Lädke**[1], **Johannes Schröder-Schetelig**[1,2,3], **Vishalini Venkatesan**[1], **Raúl A. Quiñonez Uribe**[1,3], **Claudia Richter**[1,3,4], **Rupamanjari Majumder**[1,2,3], **Stefan Luther**[1,2,3,5]*

**1** Research Group Biomedical Physics, Max Planck Institute for Dynamics and Self-Organization, Göttingen, Germany, **2** Institute of Pharmacology and Toxicology, University Medical Center Göttingen, Germany, **3** DZHK (German Center for Cardiovascular Research), Partner Site Göttingen, Germany, **4** WG Cardiovascular Optogenetics, Lab Animal Science Unit, Leibniz Institute for Primate research, Göttingen, Germany, **5** Institute for the Dynamics of Complex Systems, Göttingen University, Germany

* sayedeh.hussaini@ds.mpg.de (SH); stefan.luther@ds.mpg.de (SL)

**Data Availability Statement:** For the simulation study of this work, all relevant data are included in the manuscript and in the Supporting information files. All relevant data for the experimental study

## Abstract

Rotating spiral waves in the heart are associated with life-threatening cardiac arrhythmias such as ventricular tachycardia and fibrillation. These arrhythmias are treated by a process called defibrillation, which forces electrical resynchronization of the heart tissue by delivering a single global high-voltage shock directly to the heart. This method leads to immediate termination of spiral waves. However, this may not be the only mechanism underlying successful defibrillation, as certain scenarios have also been reported, where the arrhythmia terminated slowly, over a finite period of time. Here, we investigate the slow termination dynamics of an arrhythmia in optogenetically modified murine cardiac tissue both *in silico* and *ex vivo* during global illumination at low light intensities. Optical imaging of an intact mouse heart during a ventricular arrhythmia shows slow termination of the arrhythmia, which is due to action potential prolongation observed during the last rotation of the wave. Our numerical studies show that when the core of a spiral is illuminated, it begins to expand, pushing the spiral arm towards the inexcitable boundary of the domain, leading to termination of the spiral wave. We believe that these fundamental findings lead to a better understanding of arrhythmia dynamics during slow termination, which in turn has implications for the improvement and development of new cardiac defibrillation techniques.

## Author summary

Tachycardia and fibrillation are the most common precursors of sudden cardiac death. They are characterised by the significant increase in heart rate with or without irregular pumping, resulting in insufficient cardiac output. Research shows that high-frequency electrical spiral waves underlie these abnormal cardiac rhythms. These waves suppress the natural pacemaker of the heart to drive the overall cardiac electrical activity with poor

are available at the following link: https://doi.org/
10.5281/zenodo.10228296.

**Funding:** SL acknowledges support by the Max
Planck Society, the German Center for
Cardiovascular Research Partnersite Goettingen,
and the German Research Foundation through SFB
1002 Modulatory Units in Heart Failure. The
funders had no influence on the study design, data
collection and analysis, decision to publish, or
preparation of the manuscript.

**Competing interests:** The authors have declared
that no competing interests exist.

efficiency. Thus, restoration of sinus rhythm requires the elimination of these waves using a technique called defibrillation. Here, a single high-voltage shock forces the heart's electrical activity to undergo an instantaneous phase reset (phase is a point of the system in an oscillatory cycle). Despite high success rate, electrical defibrillation has considerable negative side effects such as intense pain, trauma, and tissue damage. These disadvantages motivate the search for low-energy alternatives to conventional defibrillation. For this to work, a deeper understanding of arrhythmia dynamics during successful termination is required.

Cardiac optogenetics opens a pathway for optical control of arrhythmia dynamics in a genetically modified tissue. Optogenetic experiments point to the existence of other slow mechanisms of successful defibrillation, which not fully understood. Exploring these mechanisms is crucial for the development of optimal defibrillation strategies to treat complex arrhythmias. Therefore, we have used a model based on cardiac optogenetics to study spiral wave termination using a single global light pulse at different light intensities and pulse lengths. We find that spiral wave termination at low light intensities occurs via a mechanism involving slow progressive dissolution of its core. Thus, we provide an explanation for the slow termination of arrhythmias at low defibrillation intensities. In addition, we have performed *ex vivo* studies in Langendorff-perfused mouse hearts controlling ventricular arrhythmias with a single global optical pulse. Optical imaging of an intact mouse heart during a ventricular arrhythmia shows slow termination of the arrhythmia, which is due to a known mechanism of action potential prolongation observed during the last rotation of the wave.

## Introduction

Excitable media are complex dynamical systems that have the ability to support the formation and sustenance of non-linear excitation patterns, such as the spiral wave (in a 2 dimensional- (2D) system) and scroll wave (in a 3 dimensional- (3D) system) [1, 2]. In the heart, the occurrence of these waves is associated with life-threatening ventricular tachyarrhythmia, rhythm disorders, which are considered to be major precursors of sudden cardiac death [3–6]. Clinically, arrhythmias are treated by removing all abnormal electrical activity from the heart, to allow the system to restore its regular function. This is most effectively achieved by electrical defibrillation. Electrical defibrillation applies a global electrical high-energy shock to the heart. Unfortunately, despite its high success rate, electrical defibrillation has considerable negative side effects such as intense pain, trauma, and tissue damage [7–12]. These disadvantages motivate the search for low-energy alternatives to conventional defibrillation [13–16]. For this to work, a deeper understanding of arrhythmia dynamics during successful termination is required. In principle, the applied electric field causes the intrinsic heterogeneities of the heart muscle to form virtual electrodes. These electrodes generate new excitation waves [17–22] that propagate intramurally and interact with the preexisting electrical abnormality. On the one hand, these interactions force synchronization of these abnormal waves at high stimulation amplitude. This leads to a high termination rate of arrhythmias. On the other hand, a decrease in stimulation amplitude leads to a lower termination rate, which could be due to the fact that the generation rate of the abnormal waves is higher than the termination rate during the wave-wave interaction. Nevertheless, it is of great interest to understand the underlying mechanism of low amplitude cardiac arrhythmia termination, as the common side effects mentioned above may be reduced.

Depending on the spatiotemporal dynamics of the abnormal electrical activity, namely the spiral wave and scroll wave within cardiac tissue, the approach to control the arrhythmia must be adapted. During conventional defibrillation, the mechanism is based on ensuring a complete phase reset of the heart's electrical activity and is therefore almost immediate. However, this may not be the only mechanism underlying successful defibrillation, as termination of arrhythmias with transient time has also been observed, numerically and experimentally, in studies of [23–25]. Therefore, understanding spiral wave dynamics and the mechanisms underlying successful defibrillation are of great interest to the nonlinear dynamics and complex excitable systems community. It is important to emphasize that such in-depth experimental investigations require powerful research tools. In addition to probes that can overcome the major challenges of tracking and visualizing scroll waves in cardiac tissue [26, 27], tools are needed that allow controlled and reversible manipulation of cellular processes.

Optogenetics is a technology that enables such control at the cell, tissue, and organ level. This technique is utilized in a vast range of study purposes, from basic [28–32] to application studies [23, 33–37]. In these works, arrhythmia in genetically modified cardiac tissue is controlled by different global and structured illumination patterns. At low light intensities, it was observed that the arrhythmia is not terminated abruptly but only after a few rotations of spiral waves. These observations prompted us to investigate in more detail the progression of arrhythmia dynamics during termination. Here, we use optogenetics to investigate the mechanisms that lead to the termination of spiral waves at different energy levels, both in *in silico* and in *ex vivo* systems. Defibrillation energy is given by regulating the intensity and duration of the applied light pulse. We find that the main mechanism for successful termination of spiral waves at high light intensities is the abrupt excitation of the entire medium containing the wave, which prevents its further propagation and results in a spontaneous termination. However, at low light intensities with a long pulse length, we observe slow termination of a spiral wave in both systems of *in silico* and *ex vivo*. In the latter case, the termination of a spiral wave rotating in an transgenic mouse heart, is proceeded by prolongation of the last action potential. However, in the former case, a progressive dissolution of the spiral wave core, in a 2D domain of ventricular mouse heart, was followed by a termination with a transient time.

## Materials and methods

### Ethics statement

All experiments in the intact mouse heart were done in accordance with the guidelines from Directive 2010/63/EU of the European Parliament on the protection of animals used for scientific purposes and the current version of the German animal welfare law and were reported to our animal welfare representatives. The experimental protocol was approved by the responsible animal welfare authority (Lower Saxony State Office for Consumer protection and Food Safety). Humane welfare-oriented procedures were carried out in accordance with the Guide for the Care and Use of Laboratory Animals and done after recommendations of the Federation of Laboratory Animal Science Associations (FELASA).

### Numerical study

For our numerical studies, we use a 2D continuum model of the membrane voltage $V$ spread across an optogenetically modified monolayer of cardiac cells. Spatiotemporal evolution of $V$

is described using a reaction-diffusion-type partial differential equation:

$$\frac{\partial V}{\partial t} = \nabla . \mathcal{D} \nabla V - \frac{I_{ion} + I_{ChR2}}{C_m} \qquad (1)$$

Here, $C_m = 1.0\ \mu F/cm^2$ is the specific capacitance of a single cell membrane, $\mathcal{D} = 0.00014\ cm^2/ms$ is the diffusion coefficient, which accounts for intercellular coupling, and is set to obtain conduction velocity of a propagating plane wave to 43.9 cm/s. We describe the total ionic current ($I_{ion}$) produced by a cell according to the formulation of [38, 39], who developed this model for adult mouse ventricular cardiomyocytes. Our simulation domain contains $100 \times 100$ grid points, which translates to a physical size of 25 mm × 25 mm.

In order to incorporate light-sensitivity to the cardiac cells we combine a 4-state model of Channelrhodopsin-2 (ChR2, a light-gated protein) [40] to the Bondarenko model [38, 39, 41]. The ChR2 current ($I_{ChR2}$) is described in Eqs (2) and (3).

$$I_{ChR2} = g_{ChR2}\, G(V)(O_1 + \gamma O_2)(V - E_{ChR2}) \qquad (2)$$

$$G(V) = \frac{\left[ \left( 10.6408 - 14.6408 \cdot exp\left( \frac{-V}{42.7671} \right) \right) \right]}{V} \qquad (3)$$

Here, $g_{ChR2}$ is the maximum conductance of the ChR2 ion channel, G(V) is the voltage-dependent rectification function. $O_1$ and $O_2$ are open state gating variables, as opposed to two closed state gating variables $C_1$ and $C_2$, that together comprise the 4-state ion channel. $C_1$ and $C_2$ are described in Ref. [40]. $\gamma$ is the ratio $O_2/O_1$, and $E_{ChR2} = 0mV$ is the reversal potential of this non-selective cation channel. In our numerical studies we considered 25 different initial conditions. The source code for our numerical simulations is provided in S1 Code.

**Calculation of core radius.** In our simulations, to calculate the core radius of the spiral wave under global illumination, we first use the Canny filter method to detect the edge, and then a circle is identified and fitted to the core area using the Hough transform method (see S2 Fig).

## Experimental study

**Experimental measurements on control of ventricular arrhythmia in the intact mouse heart.** In experiments, we report results of our *ex vivo* studies using Langendorff-perfused hearts obtained from $\alpha$-MHC-ChR2 transgenic mice (source: Dr. S. Sonntag, PolyGene AG, Switzerland). To induce arrhythmias, we apply 30 electrical pulses with an amplitude of 2.3–2.5V, width of 3–7ms, and a frequency of 30–50Hz using a needle electrode. To stabilize the arrhythmia, (*i*) the concentration of KCl in tyrode solution was reduced from 4mM to 2mM, and (*ii*), 100 μM Pinacidil, (a KATP channel activator) was added to the tyrode. In all our experiments we define arrhythmias to be persistent, if they last at least 5s. For global illumination, we placed three equidistant LEDs (wavelength of 460 nm) each with an angular separation of 120˚ around the bath and simultaneously illuminate the heart from all three directions with a single blue light pulse generated by the three LEDs [23]. In our experimental studies we used 5 different mouse hearts with 10 trials each.

**Optical mapping experiments on ventricular arrhythmia and its termination in the intact mouse heart.** In this study, we use optical mapping using potentiometric fluorescent dye to visualize the dynamics of ventricular arrhythmias in an intact optogenetic mouse heart [42]. To this end, we first introduce the contraction upcoupler blebbistatin (c = 5 μM) into the heart via perfusion. This leads to a decrease in the mechanical contraction of the heart (after about 20 minutes), which results in a significant reduction in the distortion of the fluorescence

signals [43]. To record the changes of the membrane voltage of cardiomyocytes, we stain the heart with a bolus injection of a voltage-sensitive dye: the red-shifted dye Di-4-ANBDQPQ (c = 50 μM, Thermo Fisher Scientific). During optical mapping imaging, excitation light from a 625nm mounted LED (M625L3, Thorlabs) was first filtered through a bandpass filter (FF02–628/40-25, Semrock) and then reflected through a dichroic long-pass mirror (FF685-Di02–25x36, Semrock). Then, the emitted light was first collected by a 775 ± 70nm bandpass filter (FF01–775/140-25, Semrock) and finally by the camera (EMCCD, Cascade 128+, Photo-metrics) with a spatial resolution of 64 × 64 pixels (133 μm per pixel) at 1 kHz. Optical Map-ping recordings were acquired with software MultiRecorder and analysed with software PythonAnalyser (both developed by Research Group Biomedical Physics). Electrical signals were recorded with 16-channel data acquisition system MP160 and software AcqKnowledge (BIOPAC Systems, Inc.). We observed the crosstalk between the blue stimulation light and the fluorescent dye leads to an increase in the fluorescence baseline. Therefore, to counteract this signal artifact, the optical signal is re-normalized by division to the recorded stimulation light signal.

## Results

Fig 1A illustrates a single spiral wave with a circular core trajectory, indicated by a dashed box, *in silico* study. To investigate the electrical activity within the domain inside the spiral core and in its neighborhood, we measure the membrane voltage along a line parallel to the x-axis pass-ing through the circular core (y = 12.5 mm).

Temporal evolution of *V* measured along this line is shown in the space-time plot of the membrane voltage in Fig 1B. The core region of the spiral wave shows up as a blurred area on this plot. We have outlined this region using a dashed rectangle. The information presented in Fig 1B is shown slightly differently in Fig 1C, where the voltage along the line y = 12.5 mm is reported at different time points (indicated by the color bar). We observe that during 500 ms of spiral wave rotation, there is a local minimum in the value of *V* at the spiral core. The spatial extent of the core (0.3 mm in width) is indicated using the shaded grey rectangle in Fig 1C. Note that, within the core region the value of *V* increases with time, but always remains below the excitation threshold. Thus, no activity originates from the core around which the tip of the

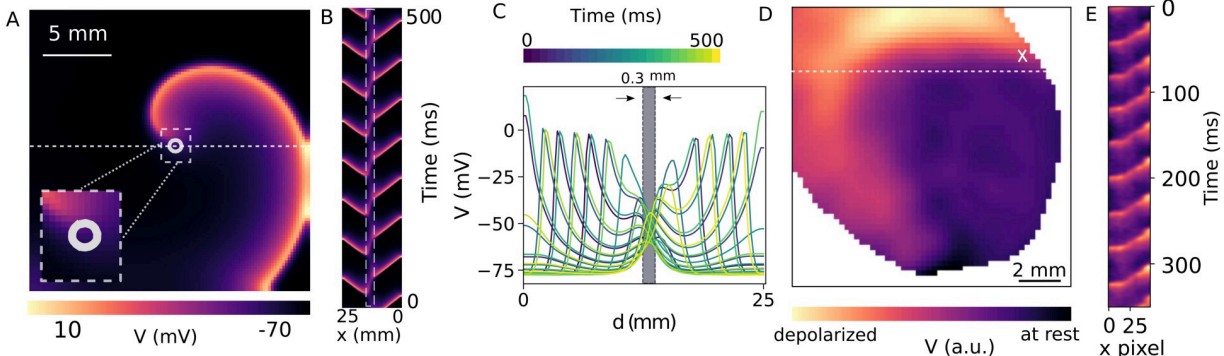

**Fig 1. Dynamics of a spiral wave *in silico* and *ex vivo*.** A) Circular tip trajectory of a spiral wave, enclosing a core of diameter 0.3 mm, in a 25 mm × 25 mm simulation domain containing mouse ventricular cardiomyocytes. B) Space-time plot of the membrane voltage along the x-axis at y = 12.5 mm, for 500 mm of spiral wave rotation. The blurred region indicated with a rectangular dashed box demonstrates the core region of the spiral wave. C) Voltage distribution during 500 ms of simulation, along the x-axis with y = 12.5 mm. The gray shaded region illustrates the electrical activities inside the core. D) Fluorescence image of a single spiral wave in an optogenetic intact mouse heart in an arrhythmic state. E) Space-time plot of the membrane voltage along the dashed line in (D), for 350 ms of spiral wave rotation.

spiral continues to rotate. Fig 1D shows a single snapshot of a fluorescence image of an optogenetic intact mouse heart in an arrhythmic state. The depolarized region shows the arm of a single spiral wave, of which the core is located in the upper right of the image and is represented by a white cross. The space-time plot of the membrane voltage during 350 ms of recording along the dashed line is shown in Fig 1E. Since the entire dynamics of the spiral wave core is not in view in this recording, the ripples in the space-time diagram show only the time evolution of $V$ of the arm of the spiral wave crossing the dashed line.

To study the dynamics of arrhythmia termination during global illumination, we apply a single optical pulse with low light intensity (LI) and long pulse length (PL) to both systems of *in silico* and *ex vivo* during arrhythmic state. Fig 2A shows termination of the spiral wave in the 2D domain during global illumination with LI of 30 μW/mm$^2$ and PL of 300 ms. We observe that the annihilation of the spiral wave occurs via gradual expansion of its core (see S1 Video). When light is applied globally to the excitable medium containing the spiral wave, the

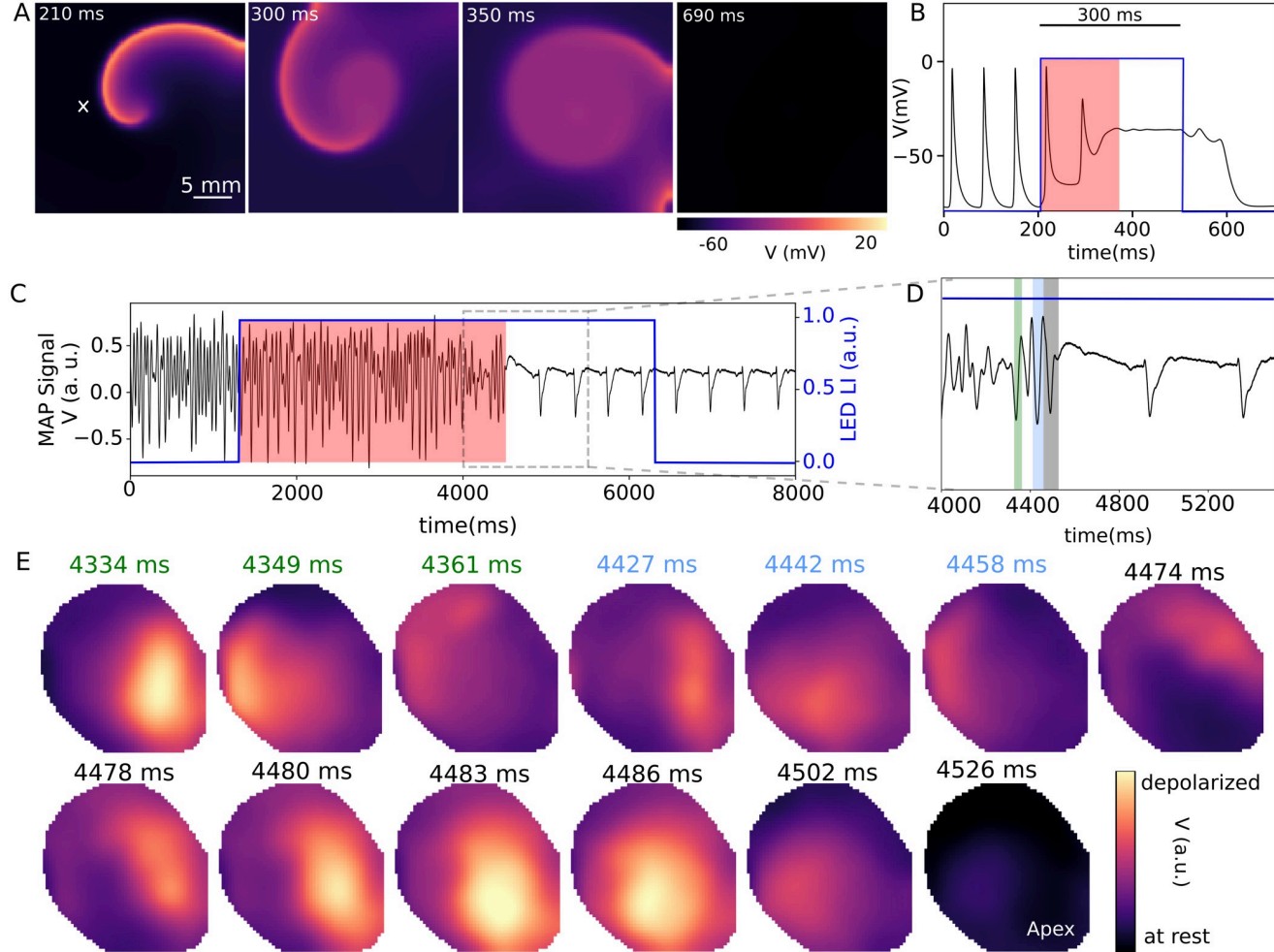

**Fig 2. Control of arrhythmia by applying a single global optical pulse with low light intensity in *in silico* and *ex vivo*.** A) Termination of a spiral wave in a two dimensional domain at light intensity (LI) of 30 μW/mm$^2$ and Pulse length (PL) of 300 ms. B) Voltage-time series of a point (shown with a cross marker in (A)) before, during, and after illumination. C) A monophasic action potential (MAP) signal (in black) of an optogenetic intact mouse heart before, during, and after illumination (in blue) at LI of 3.44 μW/mm$^2$ and PL of 5 s. D) shows a section of the MAP signal marked with a gray dashed box in (C). E) shows a series of fluorescence images of arrhythmia dynamics during illumination of a ventricular tachycardia in the mouse heart.

core region, which remains unexcited at all times by the spiral itself, becomes most susceptible to the applied excitation. Thus, the core depolarizes before any other part of the domain. By depolarizing the core we effectively convert functional reentry (free spiral wave activity) into the equivalent of an anatomical reentry, characterized by anchoring of the spiral wave around the light-induced heterogeneity at the core. At this point, the phase singularity, the unexcitable organizing center of the tip of the spiral wave, is moved into the boundary. Regions that respond next to the applied light stimulation are those that lie within the excitable gap. Thus, close to the core, subsequent excitation of the excitable gap effectively adds to the core size of the spiral, thereby contributing to its gradual expansion. Eventually, as the core expands all the way to the boundary, the rotational activity around it disappears and termination occurs (see Fig 2A for a full demonstration of the process). We refer to this mode of termination of the spiral wave, as core dissolution. The corresponding voltage time series of the recording electrode shows a very short termination transient time (the red shaded box), less than 100 ms. Thereafter, the system enters the constant elevated phase, which then decays to the resting state once the light is switched off (see Fig 2B). For *ex vivo* study we used a light stimulus with LI = 3.44 μW/mm$^2$ and PL = 5 s. Fig 2C shows a monophasic action potential (MAP) signal (in black) before, during, and after illumination. It shows the sinus rhythm returns after 3190 ms of illumination. Fig 2D presents a closer look at the MAP signal of the heart during the last few rotations of arrhythmia and the first few action potentials of sinus rhythm. Fig 2E illustrates a sequence of fluorescence frames during the rotation of the spiral wave at the time between 4334 ms-4361 ms (indicated in Fig 2D with a green shaded box on the left). It shows the wave spreads first at the apex and then continues rotating clockwise within approximately 40 ms. The next three fluorescence imaging frames show the second last rotation of arrhythmia during 4427 ms to 4458 ms indicated in Fig 2D with a blue shaded box. The fluorescence frames from 4474 ms to 4526 ms illustrate the dynamics of the arrhythmia during the last rotation marked by a grey shaded box in Fig 2D. During this last rotation of the spiral wave the increase of the wavelength of the wave can be seen. This leads to a gradual propagation of the excitation wave (which is no longer a spiral wave at the time $\geq$ 4483 ms) over the entire area. Between 4483 ms and 4486 ms, the entire area is almost in the excited state and then falls back into the resting state at the time of $\approx$ 4526 ms (see S2 Video).

Comparing the transient termination time in the 2D numerical simulation and in the *ex vivo* experiment, shown as a red shaded area in Fig 2B and 2C, it can be seen that the transient time is relatively long in the case of the experiment. This could be due to the very different systems: a 2D cardiac tissue in which a single spiral wave rotates in a homogeneous single layer of cardiac tissue versus an intact mouse heart in which a spiral wave propagates in an inhomogeneous cardiac tissue with a more complex geometry. Moreover, light attenuation through the cardiac tissue in an intact heart enhances the complex spatiotemporal dynamics of the wave in the system.

Our numerical results indicate that spiral wave termination via progressive dissolution of the core can be observed that at low LIs, whereas, at high LIs, termination occurs abruptly, similar to conventional electrical defibrillation. To develop a deeper understanding of the process of core dissolution, we apply a single global pulse to the spiral wave in a wide range of LIs from sub-threshold illumination (no excitation wave is triggered in this range of LIs) to supra-threshold illumination (an excitation wave is triggered in this range of LIs) with the PL of 500 ms. Fig 3A shows a space-time plot of the membrane voltage along a line in the range (shown in Fig 1A) from sub-threshold LIs, $\leqslant$ 20 μW/mm$^2$ to supra-threshold LIs, $\geqslant$ 25 μW/mm$^2$. We observe that for the sub-threshold illumination, increasing the LI increases the period of the spiral wave ($T_s$), which leads to a decrease in its conduction velocity [30]. In the supra-threshold régime, at low LIs (between 25–50 μW/mm$^2$) an excitation wave is created at the core

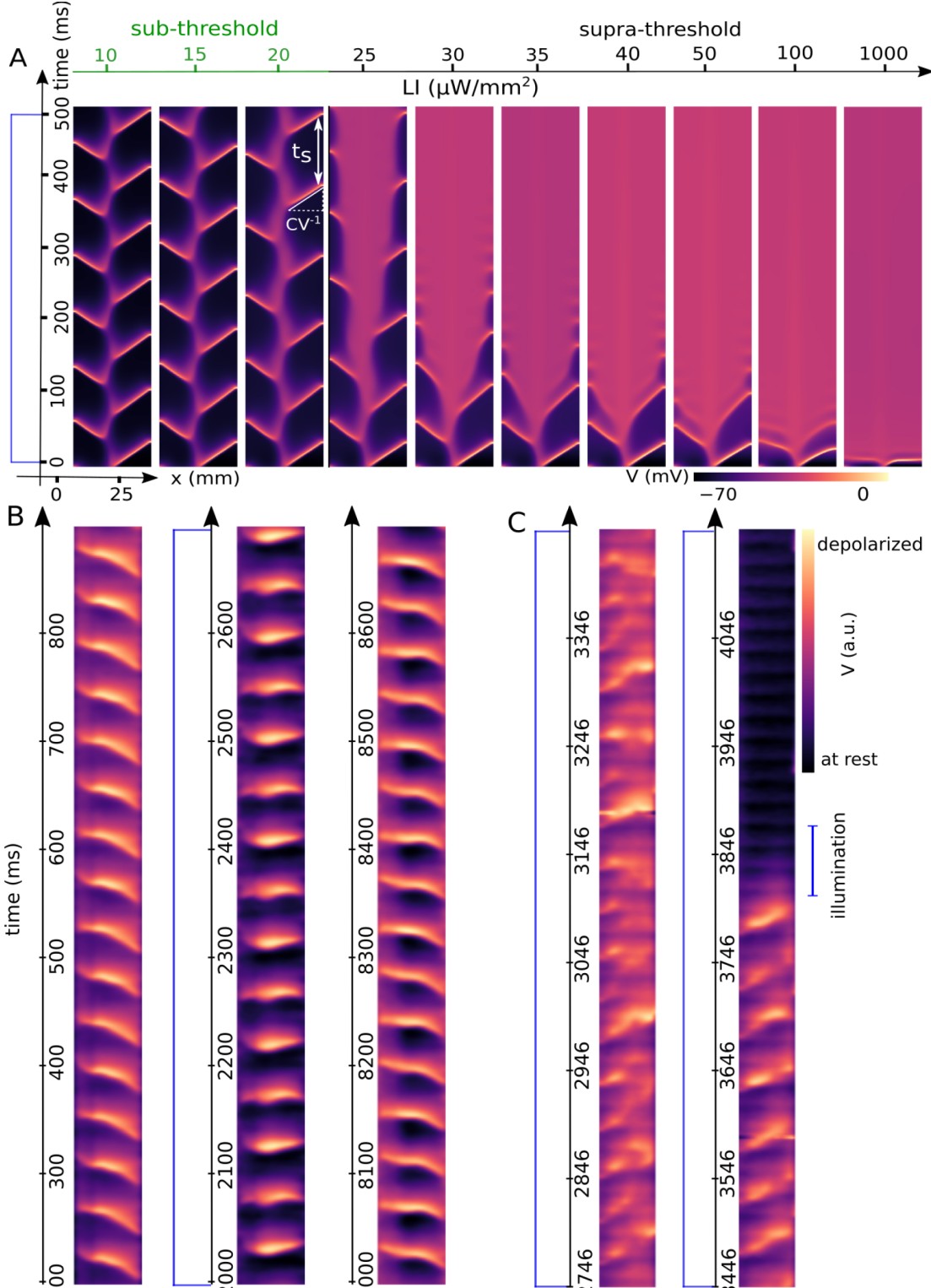

**Fig 3. Non-spontaneous termination of a spiral wave *in silico* and *ex vivo*.** A) Space-time plots of the membrane voltage along the x-axis at y = 12.5 mm, in a two dimensional domain with an existing spiral wave during 500 ms of a single pulse global illumination. B) Space-time plot of membrane voltage of an intact mouse heart in an arrhythmic state before, during, and after illumination. C) Space-time plot of the membrane voltage of an intact mouse heart showing arrhythmia is terminated during illumination.

region which propagates through the whole domain and excites it to an elevated constant potential. Increasing LI leads to faster initiation of the excitation wave, reaching the elevated constant potential at shorter transient times. At very high LIs, ($\geqslant 80\ \mu W/mm^2$) the system immediately enters the elevated constant potential when the light is switched on. A space-time plot of the membrane voltage of an intact mouse heart during 8500 ms sections of optical mapping recording is shown in Fig 3B. It shows that the arrhythmia consists of a single spiral wave with a stationary core dynamic and a rotational frequency of $\approx 20$ Hz before illumination. During a single global optical pulse with a PL of 5000 ms and an LI of 3.21 $\mu W/mm^2$, the frequency of the arrhythmia decreases to $\approx 18.75$ Hz. It is important to note that the frequency reverses its original value of $\approx 20$ Hz when the light is turned off. Fig 3C demonstrates a space-time plot of the membrane voltage of the same optical mapping recording as shown in Fig 2E. Here a single global optical illumination with LI of 3.44 $\mu W/mm^2$ and PL of $\approx 5$ s is applied to the mouse heart. It illustrates that during the first $\approx 4000$ ms the arrhythmia dynamics consists of a single spiral wave with a non-stationary meandering core dynamics. Then, during the last nine rotations of the spiral wave, the wave propagates with a stationary core dynamic. Finally, the arrhythmia is terminated at $\approx 4500$ ms due to the increase in action potential duration, and the first planar excitation wave of sinus rhythm propagates at $\approx 4900$ ms.

To investigate the transition dynamics into and out of the core dissolution process, we apply a shorter light stimulus PL = 150 ms, with simulation time of 300 ms, to the same range of LIs. We observe that at subthreshold LI, removal of the light pulse restores the system to its original state (see Fig 4A). At LI = 25 $\mu W/mm^2$ and 30 $\mu W/mm^2$ core dissolution stops when the light is turned off. At LI = 25 $\mu W/mm^2$, the spiral wave continues to rotate at the same location within the domain. However, at LI = 30 $\mu W/mm^2$, the new core position is shifted relative to the original state. At LI $\geqslant 35\ \mu W/mm^2$, the core dissolution continues and leads to the termination of the spiral wave. Fig 4B shows radius growth rate of the spiral wave's core when global uniform illumination is applied at three different LI of 25, 30, and 35 $\mu W/mm^2$. The growth rate was measured with the time interval of 10 ms. The slope of each fitted line shows an increase with increasing LI. The details of the slopes and intercept of each fitted line for the different LI of 25, 30, 35, 40, and 50 $\mu W/mm^2$ are shown in Table 1. We also measured the minimum illumination time (PL$_{critical}$) to terminate the spiral wave at different LIs of 25–50 $\mu W/mm^2$. Fig 4C shows the decrease in PL$_{critical}$ with increase in LI, in which the PL$_{critical}$ of 498, 190, 100, 100, and 80 ms corresponds to LIs of 25, 30, 35, 40, 45, and 50 $\mu W/mm^2$, respectively.

Next, to control the dynamics of the spiral wave in favor of termination, we apply a single global optical pulse to the pattern in Fig 1A and 1D at different LIs and PLs. Then, we compare the dose-response curves for arrhythmia termination in experiments and simulations. Dose-response diagram for optical stimulation displayed in Fig 5A and 5C shows increasing LI and PL, both independently and together, lead to an increase in arrhythmia termination efficiency. To calculate the termination time (both in the simulation and in the *ex vivo* experiment), the time between the switching on of the light and the last peak of the arrhythmia before termination is considered. The time required for the successful termination of arrhythmias with PL = 500 ms demonstrated in Fig 5B and 5D shows that the transient time is longer than 100 ms at low LIs (50 $\mu W/mm^2$) in both numerical simulations and experiments. At high LIs ($\geqslant$ 80 $\mu W/mm^2$), due to the abrupt annihilation of the spiral, the termination time is much shorter and varies in a narrow range of 10 to 20 ms. These results are presented in Fig 5D. The transient termination time of the experimental data shown in Fig 5B is in qualitative agreement with the numerical study (Fig 5A). Although the two systems (2D domain and intact mouse heart) are very different from each other, the similar trends in termination transient time suggest that the two systems must share a common dynamical feature. The kinetics of

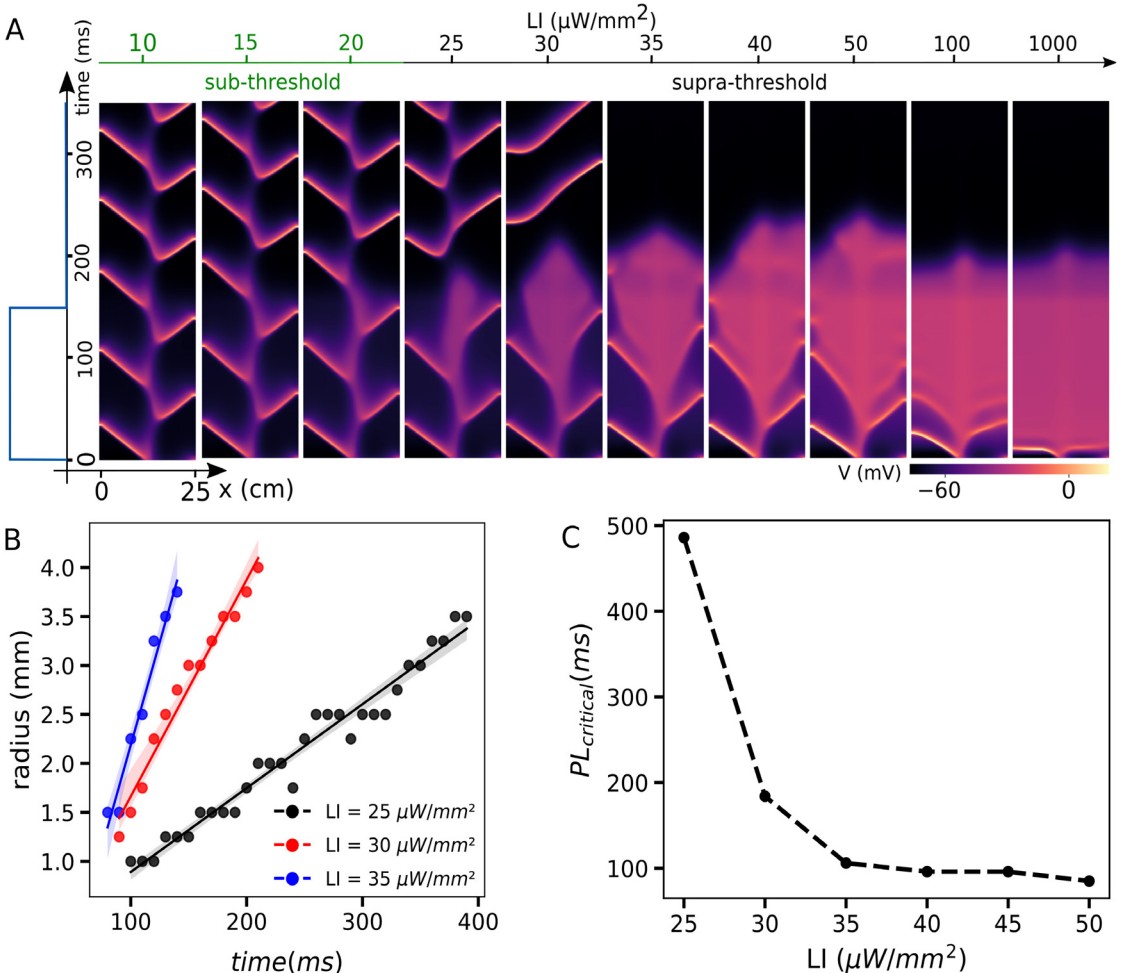

**Fig 4. Dynamics of a single spiral wave *in silico* study.** A) Shows space-time plot of the membrane voltage along the x-axis at y = 12.5 mm, in a two dimensional domain with an existing spiral wave. A global pulse illumination with pulse length (PL) of 150 ms (shown with a blue trace on the left) was applied at different light intensities (LIs) during 350 ms simulation time. B) Shows the core size growth of the spiral wave with the time interval of 10 ms during global illumination at different LIs. C) Minimum PL ($PL_{critical}$) required to terminate the single spiral wave at various LIs.

Channelrhodopsin-2 (ChR2) could possibly explain the similarity between numerics and experimental results. The activation of this protein leads to three different phases: 1) an initial peak of the incoming cation current ($I_{peak}$), 2) decay of this peak to a constant phase with the width of the optical pulse duration ($I_{cnst}$), and 3) decay of the current to the baseline when the

**Table 1. Comparison between the slopes and intercepts of the radius growth of the core of the spiral wave during global illumination at different LIs.**

| LI (µW/mm²) | Slope | Intercept |
|---|---|---|
| 25 | 0.0085 | 0.034 |
| 30 | 0.022 | -0.54 |
| 35 | 0.04 | -2.0089 |
| 40 | 0.038 | -1.3 |
| 50 | 0.045 | -1.3 |

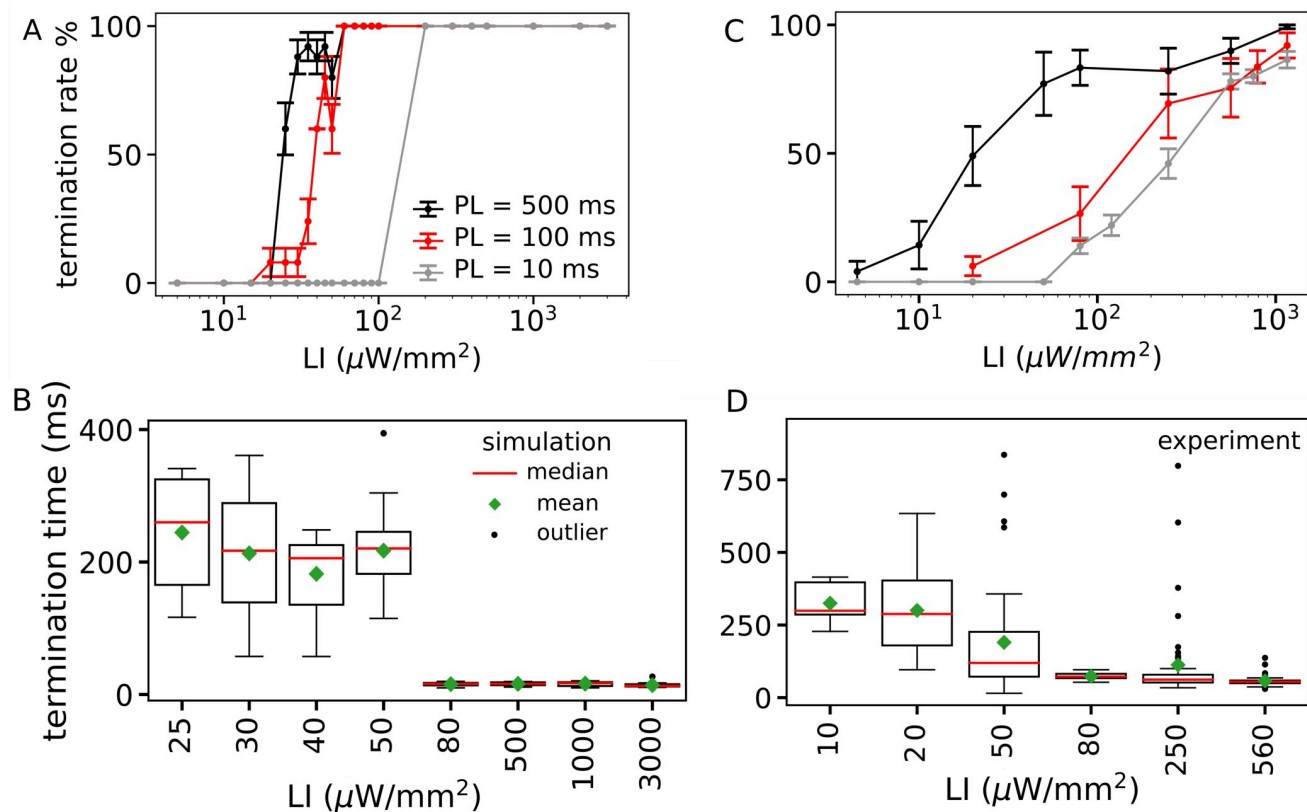

**Fig 5. Termination of arrhythmia applying a single pulse global illumination *in silico* and *ex vivo*.** A) Termination rate of a single spiral wave in a two dimensional domain with size of 25 mm × 25 mm simulation (N = 25) at different light intensities (LIs) and pulse lengths (PLs) (percentage of successful attempts reported as mean ± SEM). B) Termination time at different LIs with PL of 300 ms. C) Arrhythmia termination rate vs. LI in Langendorff-perfused intact mouse hearts (N = 5) for different PLs. D) Termination time at different LIs and PL of 500 ms.

light is switched off. Fig 1, supplementary, illustrates the kinetics of ChR2 current and corresponding membrane voltage activity at different LIs. It shows the application of sub-threshold LIs leads to a small inward current of $I_{ChR2}$ with low amplitude of $I_{peak}$ and $I_{cnst}$. This leads to an increase in membrane voltage below the excitation threshold (indicated by the dashed line). However, the application of supra-threshold illumination ($\geqslant 25$ µW/mm$^2$) causes the large inward current of $I_{ChR2}$ to exceed the excitation threshold and an excitation activity called action potential occurs. The application of low-LI supra-threshold stimuli (e.g. 25 and 30 µW/mm$^2$ in S1 Fig) results in slow activation of this channel with low $I_{peak}$ and $I_{cnst}$, so that the arrhythmia can be terminated with a longer optical PL. This leads to a transient termination time long enough to include a few turns of the spiral wave. On the other hand, high-LI supra-threshold stimuli (e.g. 100 and 1000 µW/mm$^2$ in S1 Fig) cause immediate activation of the ion channel, which results in the flow of a large current $I_{peak}$ in a very short time. This leads to immediate excitation of the heart tissue, which can lead to an abrupt termination without a long transient time.

## Discussion

In this work, we investigate the mechanisms of arrhythmia termination during global illumination at low light intensities with a long pulse length. Our *in silico* studies on a 2D domain of

ventricle mouse heart show arrhythmia termination with a transient time. During this time the core of the spiral wave is excited upon global illumination and leads to the dissolution of the core. During this process the excited region grows over time and pushes the arm of the spiral wave to the boundary, leading to its termination. In *ex vivo* studies on the intact optogenetic mouse heart, we have visualised arrhythmia termination with optical mapping technique. For a case of arrhythmia termination with a transient time, we observe an increase in the action potential during the last rotation of the spiral wave. This leads to a larger excitation of the heart surface so that the spiral wave cannot propagate further, resulting in its termination.

In cardiac optogenetics, numerous studies have been carried out to control cardiac arrhythmias in murine hearts with different lighting patterns, such as structured [33, 34, 37] and global [23]. Uribe *et al.* applied global illumination to control cardiac arrhythmias in intact transgenic mouse hearts at different PLs and LIs [23]. They showed for some successful termination cases that there is a non-spontaneous termination at low LIs with a long PL of 1000 ms. Our numerical studies suggest that core dissolution may be the underlying mechanism leading to the slow termination that they have observed for the long illumination pulse with low-intensity. In another numerical study, we demonstrate that drift, a spatial translation of the spiral wave core along the illumination gradient, may underlie optogenetics defibrillation [30]. Dillon *et al.* provided optical mapping data showing the effect of electrical stimulation on the increase in action potential duration (APD) with simultaneous extension of the wavelength of the travelling wave [44]. With increase of APD, the excitable region for the spiral wave to propagate is reduced [24, 25] which may lead to a termination with transient time. Biasci *et al.* showed an incremental effect on APD by applying global sub-threshold illumination to a propagating planar wave in the domain [45]. All these studies show that the combination of different mechanisms can lead to the arrhythmia extinction at low-amplitude stimulation, or that occasionally one plays a dominant role over the others in causing termination of arrhythmias.

In this work, the experimental studies were performed on intact mouse hearts, the geometry and structural features were completely different from the homogeneous 2D numerical system. Unlike 2D, in 3D *ex vivo* (or anatomical geometries for that matter), wave dynamics is determined not just by the area of excitable surface on the heart, but also by the depth of excitable tissue beneath the surface, which allows scroll filaments (a line of phase singularities in 3D) to resist removal by surface synchronization. Thus, arrhythmia termination occurs slowly in 3D, compared to 2D. This may explain why in experiments we see a smoother distribution for arrhythmia termination, than in 2D, where the dependence is rather abrupt.

From a clinical perspective, it is important to understand the possible mechanisms of arrhythmia termination in thick cardiac tissue (e.g, from the left ventricular wall), which is 3D. Previous studies have shown that depending on the thickness of the tissue, a scroll filament can bend, twist, break up, grow and/or shrink inside the bulk of the tissue [46]. Presence of inhomogeneities inside the tissue further add to the complexity of the wave dynamics in that, they allow attachment/detachment of the filaments to/from their locations, as well as promote the formation of 'seed waves' which have the potential to regenerate a full scroll [47]. These dynamical behaviours can also occur under the influence of illumination on the surface of cardiac tissue during an arrhythmia. Depending on the true depth of penetration of the applied light, one can expect the tissue substrate to behave inhomogeneously. Thus, it would be interesting to observe how the proposed mechanism of slow termination of a scroll wave would take effect in such a scenario. We conjecture that the proposed mechanism of slow termination can then be seen to occur when the fibre orientation is mostly homogeneous, does not show sharp changes and light penetration is sufficiently deep.

## Conclusion

Various studies have investigated and postulated different mechanisms underlying optogenetic defibrillation [23, 30, 48]. In this work, we investigate the mechanisms of arrhythmia termination during global illumination at very low supra-threshold light intensities with a long pulse length. Our *in silico* studies on two-dimensional domain of ventricle mouse heart show a slow termination of arrhythmia in which the core of the spiral wave is depolarized during global illumination, leading to its dissolution. During this process, which we refer to as core dissolution, the depolarized region grows over time and pushes the arm of the spiral wave to the boundary, leading to its termination. In *ex vivo* studies on the intact mouse heart, we have visualised arrhythmia termination with optical mapping technique. When arrhythmia termination is not instantaneous, we observe an increase in the action potential during the last rotation of the spiral wave. This leads to a larger depolarisation of the heart surface so that the spiral wave cannot propagate further, resulting in its termination. This work provides fundamental findings which could have implications for the improvement and development of new cardiac defibrillation techniques.

## Supporting information

**S1 Code. The source code used to obtain the numerical simulations reported in this manuscript.**
(ZIP)

**S1 Fig. Kinetics of ChR2 and action potential of a single cell during sub- and supra-threshold illumination.** The application of sub-threshold illumination (LIs of 10 and 20 $\mu W/mm^2$) leads to the production of a small amount of inward $I_{ChR2}$ and elevation of the membrane voltage below the excitation threshold. The application of supra-threshold illumination at low LIs (25 and 30 $\mu W/mm^2$) leads to the slow activation of the channel and in turn depolarization of the cell membrane. The application supra-threshold illumination with high LIs (100 and 1000 $\mu W/mm^2$) results in the fast activation of the channel and depolarization of the membrane voltage to more positive values.
(EPS)

**S2 Fig. Calculation of core size.** A) Shows a detected edge (white) using the Canny filter technique and a fitted circle (red) using the Hough transform method. B) illustrates the fitted circle on the snapshot of the spiral wave at a time step of 350 ms during illumination with a light intensity of 30 $\mu W/mm^2$.
(EPS)

**S1 Video.** *In silico* **study of controlling a single spiral wave in a 2D domain by applying global illumination with LI of 30 $\mu W/mm^2$.**
(AVI)

**S2 Video.** *Ex vivo* **study of controlling a single spiral wave in an intact optogenetic mouse heart by applying global illumination with LI of 3.44 $\mu W/mm^2$.**
(MP4)

## Acknowledgments

We thank all the members of biomedical physics group of Max Planck Institute for Dynamics and Self-Organization for their fruitful input. Our special thanks go to Marion Kunze, Tina Althaus, Andreas Barthel, and Laura Diaz for their technical support.

## Author Contributions

**Conceptualization:** Sayedeh Hussaini, Rupamanjari Majumder, Stefan Luther.

**Data curation:** Stefan Luther.

**Formal analysis:** Sayedeh Hussaini, Sarah L. Lädke, Vishalini Venkatesan.

**Funding acquisition:** Stefan Luther.

**Investigation:** Sayedeh Hussaini, Sarah L. Lädke, Johannes Schröder-Schetelig, Vishalini Venkatesan, Raúl A. Quiñonez Uribe.

**Project administration:** Stefan Luther.

**Software:** Sayedeh Hussaini, Sarah L. Lädke, Johannes Schröder-Schetelig.

**Supervision:** Sayedeh Hussaini, Johannes Schröder-Schetelig, Raúl A. Quiñonez Uribe, Claudia Richter, Stefan Luther.

**Visualization:** Sayedeh Hussaini, Sarah L. Lädke, Stefan Luther.

**Writing – original draft:** Sayedeh Hussaini.

**Writing – review & editing:** Johannes Schröder-Schetelig, Claudia Richter, Rupamanjari Majumder, Stefan Luther.

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
