## [Decision Letter · Decision Letter 0]

28 Jul 2023

Dear Dr. Hussaini,

Thank you very much for submitting your manuscript "A mechanism underlying slow termination of arrhythmias" for consideration at PLOS Computational Biology. As with all papers reviewed by the journal, your manuscript was reviewed by members of the editorial board and by several independent reviewers. The reviewers appreciated the attention to an important topic. They raise points related to particular aspects of the model, including mechanisms and expected results for 3D simulations, and provide comments for refining the text to improve clarity. Based on the reviews, we are likely to accept this manuscript for publication, providing that you modify the manuscript according to the review recommendations.

Sincerely,

Steven A. Niederer

Guest Editor

PLOS Computational Biology

Stacey Finley

Section Editor

PLOS Computational Biology

Reviewer's Responses to Questions

**Comments to the Authors:**

Reviewer #1: This study by Hussaini et al present a very detailed, well-written and comprehensive combined simulation and experimental study demonstrating a novel mechanism of arrhythmia termination through optical defibrillation. Throughout the comparison between simulations and experiments is strong, which significantly strengthens the study. I have only relatively minor comments.

1. The 2D vs 3D issue is very important. I can understand why the simulations were conducted in 2D, though. The discussion of these issues, however, in the Discussion nicely explains the possible reasons for the discrepancies. In future works, I would urge the authors to consider 3D simulation approaches, along with corresponding models of the decay of the exciting light into the tissue depth. I fear that, whilst this and similar works have shown great promise in 2D simulations and very thin (small mammal) hearts, it remains unclear to me how these effects would translate into a 10mm+ human left ventricle in which transmural excitation will be a significant challenge. I feel that more of a comment in this regard is important to include in the Discussion, in terms of depth-penetration and clinical translation. In particular, as well as the depth of light penetration issue, the dynamics of 3D scroll waves are more complex, particularly with thicker walls. Please comment on the potential implications of this.

2. In the conclusions, it states that in the simulations the main mechanisms of defibrillation was “pushing the arm of the spiral wave to the boundary, leading to its termination”. In the 2D square model, this has well-defined artificial barriers, which might lead to an artificially-higher termination rate than could be expected with a more continuous ‘wrap-around’ structure, like the heart. Please could the authors comment on this?

3. Related to this, it does not appear that this major mechanism (pushing the arm to the boundary) was witnessed in the experimental mouse data. Please could you clarify? Here, it seems that the main mechanism was through prolongation of the APD. Was this mechanism also seen in the simulations?

4. In the 2D model, conduction appears isotropic. Please comment on the potential effects of tissue anisotropy.

5. It is not clear whether the arrhythmias induced are VT-like or VF-like(?) Fig 1 implies that these were single spiral waves, and thus more like rapid VT rather than the more chaotic fibrillation patterns. The supplemental movies also seem to show far more monomorphic VT-like patterns. In the experiments, only frequencies are quoted. In real defibrillation cases, VF would be the primary arrhythmia which would be treated, and which would have significantly more ‘chaotic’ wavefront patterns than analysed here. I believe that this might significantly implicate some of the mechanisms you present here (for example, the ‘pushing of the spiral wave’). Please could the authors comment on this important issue.

Reviewer #2: This is a manuscript from Stephan Luther's lab, one of the world's leading experts in cardiac dynamics. The lab is renowned for using sophisticated modeling and experimental approaches to explore new low-energy defibrillator strategies. In this study, they conducted in silico and ex vivo investigations to dissect the basic mechanisms of single rotor termination during sub- and super-threshold optogenetic manipulations. The topic is very timely, and the experiments and modeling appear to be well-performed. I have just a few minor comments to improve the solidity, readability, and general interest of the manuscript:

1. Although generating more complex dynamics in a mouse heart is very challenging (and likely physically impossible), multiple coexisting rotors could be easily (?) generated in silico. Therefore, the authors could explore if the core-expansion mechanism represents a predominant termination mechanism in this more realistic scenario as well.

2. Considering that the VSD used in this experiment should absorb at 470nm, leading to an increase in the fluorescent baseline, I expect that authors have to re-normalize the "voltage" map during optogenetic illumination. If this is the case, it should be stated in the text.

3. Figure 2D and Figure 2B in the text seem to refer to incorrect panels.

4. The use of two different LI scales in Figure 5 can reduce readability. Consider homogenizing the scale to improve clarity.

5. In addition to references 17 - 19, two additional works should be cited in relation to lighting patterns illumination: doi.org/10.1038/srep35628;
doi.org/10.1113/JP276283.

**Have the authors made all data and (if applicable) computational code underlying the findings in their manuscript fully available?**

Reviewer #1: Yes

Reviewer #2: None

PLOS authors have the option to publish the peer review history of their article (what does this mean?). If published, this will include your full peer review and any attached files.

Reviewer #1: No

Reviewer #2: No

Figure Files:

Data Requirements:

Reproducibility:

References:

---

## [Editor Report · Decision Letter 1]

4 Nov 2023

Dear Dr. Hussaini,

We are pleased to inform you that your manuscript 'Dissolution of Spiral Wave's Core Using Cardiac Optogenetics.' has been provisionally accepted for publication in PLOS Computational Biology.

Best regards,

Steven A. Niederer

Guest Editor

PLOS Computational Biology

Stacey Finley

Section Editor

PLOS Computational Biology

---

## [Editor Report · Acceptance letter]

1 Dec 2023

PCOMPBIOL-D-23-00705R1 

Dissolution of Spiral Wave's Core Using Cardiac Optogenetics.

Dear Dr Hussaini,

I am pleased to inform you that your manuscript has been formally accepted for publication in PLOS Computational Biology. Your manuscript is now with our production department and you will be notified of the publication date in due course.

With kind regards,

Zsofia Freund
